# Novel target identification towards drug repurposing based on biological activity profiles

Binghan Xue[1], Yanji Xu[1], Ruili Huang[2], Qian Zhu[2]*

1 Division of Rare Disease Research Innovation, National Center for Advancing Translational Sciences (NCATS), National Institutes of Health (NIH), Rockville, Maryland, United States of America, 2 Division of Preclinical Innovation, National Center for Advancing Translational Sciences (NCATS), National Institutes of Health (NIH), Rockville, Maryland, United States of America

* qian.zhu@nih.gov

## Abstract

Rare diseases affect more than 30 million individuals, with the majority facing limited treatment options, elevating the urgency to innovative therapeutic solutions. Addressing these medical challenges necessitates an exploration of novel treatment modalities. Among these, drug repurposing emerges as a promising avenue, offering both potential and risk mitigation. To achieve this goal, we primarily focused on developing predictive models that harness cutting-edge computational techniques to uncover latent relationships between gene targets and chemical compounds towards drug repurposing. Building upon our previous investigation, where we successfully identified gene targets for compounds from the Tox21 in vitro assays, our endeavor expanded to a systematic prediction of potential targets for drug repurposing employing machine learning models built on diverse algorithms such as Support Vector Classifier, K-Nearest Neighbors, Random Forest, and Extreme Gradient Boosting. These models were trained on comprehensive biological activity profile data to predict the relationship between 143 gene targets and over 6000 compounds. Our models demonstrated high accuracy (>0.75), with predictions further validated by using public experimental datasets. Furthermore, several findings were evaluated via case studies. By elucidating these connections, we aim to streamline the drug repurposing process, ultimately catalyzing the discovery of more effective therapeutic interventions for rare diseases.

## Introduction

The concept of drug repurposing has emerged as a beacon of hope, offering a cost-effective and time-efficient approach to addressing the complexities of drug discovery [1–4]. At its core, drug repurposing involves the exploration of alternative applications for already approved or investigational drugs, tapping into their potential

**Data availability statement:** All relevant data are within the manuscript and its Supporting Information files.

beyond their originally intended uses. This strategy not only capitalizes on existing pharmacological knowledge but also expedites the drug development process by bypassing certain stages of preclinical and clinical trials [5]. A compelling rationale for promoting drug repurposing lies in the interconnected nature of disease mechanisms. Scientific evidence suggests that a single molecular target implicated in a specific disease often exerts influence on various genetic pathways associated with other rare diseases [6,7]. Therefore, unraveling the intricate relationship between chemical compounds and gene targets assumes paramount importance in the quest for breakthroughs in rare disease treatment.

Researchers employ a multifaceted approach to discover the complicated relationship between chemical compounds and gene targets, leveraging a combination of experimental techniques and computational methodologies. Experimental studies often involve high-throughput screening assays, where thousands of compounds are tested against various biological targets to identify potential interactions [8–10]. Concurrently, advanced computational algorithms, such as molecular docking and network analysis, play a crucial role in predicting the binding affinity and specificity of compounds to target proteins [11–15]. These computational models rely on structural and functional data of both compounds and target proteins, allowing researchers to elucidate the molecular mechanisms underlying drug-target interactions.

However, traditional methods like molecular docking and network analysis, while valuable, have certain limitations in the exploration of chemical compounds and gene targets. Molecular docking, for instance, relies heavily on structural data of target proteins and ligands, often overlooking the dynamic nature of protein-ligand interactions and the influence of solvent effects. This can lead to inaccuracies in predicting binding affinities and may not capture the full spectrum of interactions within complex biological systems [15–17]. Similarly, network analysis, while useful for identifying functional relationships between genes and proteins, may struggle to integrate diverse types of data and capture the dynamic nature of biological networks [18]. Additionally, traditional methods often require manual curation and parameter tuning, which is time-consuming and potentially biased [15,18–20]. Moreover, these methods may struggle with the scalability required to analyze large-scale datasets, limiting their applicability in the era of big data [21,22]. These inherent limitations underscore the need for complementary approaches, such as machine learning and artificial intelligence, to overcome these challenges and unlock new insights in drug discovery and molecular biology.

In recent years, machine learning (ML) and artificial intelligence (AI) tools have revolutionized the exploration of the intricate relationship between chemical compounds and gene targets. These tools enable researchers to analyze vast datasets comprising chemical structures, biological activities, and genetic information to uncover novel associations and predict potential interactions [23–30]. ML algorithms, such as Support Vector Classifier (SVC), Random Forests, and deep learning models, are trained on high-dimensional data to classify compounds based on their biological activities or to predict the binding affinity of compounds to specific target proteins [31–34]. Moreover, different AI and ML tools have been widely used for drug

safety evaluation offering beneficial information for drug repurposing [35–37]. Application of advanced machine learning approaches such as k-nearest neighbors (KNN) [38], SVC [39], and extreme gradient boosting (XGB) [40], in drug repurposing exhibited high predictive performance [41,42]. Integrating these computational techniques with experimental validation can accelerate the drug discovery process and offer insights into the mechanisms of action underlying therapeutic effects.

The Tox21 dataset is a pivotal resource in the domains of predictive toxicology and drug discovery [41,43,44]. The Toxicology in the 21st Century program, Tox21, is a collaborative effort between the National Institutes of Health (NIH), the U.S. Environmental Protection Agency (EPA), and the U.S. Food and Drug Administration (FDA) [45–48]. The Tox21 dataset encompasses a plethora of biological activity information derived from screening a collection of ~10,000 drugs and environmental chemicals (Tox21 10K compound library) against a panel of *in vitro* cell-based and biochemical assays, addressing a wide spectrum of biological targets and pathways pertinent to toxicology [48–50]. The inherent advantages of the Tox21 dataset in predictive modeling for drug repurposing are rooted in its extensive scope and diversity. These characteristics facilitate the development of robust machine learning models capable of forecasting the toxicity and potential adverse effects of various chemicals and pharmaceuticals. Harnessing the vast repository of data within Tox21 offers the prospect of expediting the identification of safe and efficacious drug candidates, thereby streamlining the drug discovery process and mitigating reliance on resource-intensive experimental assays.

In this investigation, we compiled gene targets found to be significantly associated with Tox21 chemicals from our previous study [51] and developed predictive models employing four distinct ML algorithms aimed at uncovering novel gene targets associated with specific drugs. Our analysis revealed previously unrecognized gene-drug pairs, which presents opportunities for further exploration in clinical settings, thus facilitating drug repurposing endeavors across a diverse range of medical conditions.

## Materials and methods

### Data preparation

**Tox21 data preparation.** The Tox21 10K compound library contains around 10,000 substances, of which 8,971 are distinct entities, covering a broad spectrum of categories including drugs, pesticides, consumer products, food additives, industrial chemicals, and cosmetics [52]. In this study, we employed quantitative high-throughput screening (qHTS) data obtained from screening the Tox21 10K library against 78 in vitro assays. Detailed assay data are accessible from the public Tox21 website (https://tripod.nih.gov/pubdata/). Compound activity was measured by the curve rank metric, which ranges from -9–9 and is determined by various attributes of the primary concentration-response curve, including potency, efficacy, and quality. A notably positive curve rank indicates robust activation, whereas a large negative curve rank signifies potent inhibition of the assay target. Examples of compound activity scores shown in terms of curve rank are shown in Table 1. Structure ID represents CAS Registry Number of each compound. The rest column name shows Tox21 assay name. Among 8,971 substances in the original dataset, 7,170 possessed curve rank data across all Tox21 in vitro bioassays, and only compounds with available activity data were included in subsequent analyses.

**Enriched gene target selection.** From the previous study, 7,170 compounds in the Tox21 10K library were clustered based on similarity in their activity profiles across the Tox21 *in vitro* assays resulting in 129 clusters [51]. Gene enrichment analysis was performed on each cluster yielding a total of 737 enriched gene targets. For our models to discern patterns effectively and make accurate predictions, it's important that each target has a sufficient number of compounds known to be associated with them. Hence, we tallied the number of associated drugs for each enriched gene target, selecting only those linked with at least 10 different compounds for our models. This selection process enables us to enhance the predictive capacity and significance of our subsequent analyses. The gene targets selected for this study were prioritized based on their significant enrichment in compound activity profiles derived from the Tox21 dataset. This enrichment aligns with their known involvement in key disease pathways, particularly those implicated in rare diseases. For instance,

**Table 1. Example of compound activity scores in Tox 21 dataset.**

| Structure ID | tox21-ache-p1_ ratio | tox21-ache-p3_ ratio | tox21-ahr-p1_ ratio | tox21-ahr-p1_ viability | tox21-ap1-agonist-p1_ch1 | tox21-ap1-agonist-p1_ch2 | tox21-ap1-agonist-p1_ratio |
|---|---|---|---|---|---|---|---|
| 97612-24-3 | 0 | 0 | 0.667 | -0.667 | 0 | 2 | 1.333 |
| 207801-27-2 | 0 | 0 | 3.667 | 0.667 | 0 | 0 | 0 |
| 7287-19-6 | 0 | 0 | 0.667 | 0 | 0 | 0.333 | 0.333 |
| 16323-43-6 | 0 | 0 | 0 | 0 | 0 | 0 | 0 |
| 1444-64-0 | 0 | 0 | 0 | 0 | 0 | 0 | 0 |
| 183321-74-6 | 0 | 0 | 7.667 | -5 | 0 | -2 | -0.667 |
| 2404-44-6 | 0 | 0 | 0 | 0 | 0 | 0 | 0 |
| 439-14-5 | 0 | 0 | 0 | 0 | 0 | 0 | 0 |
| 1031-07-8 | -5 | 0 | 0 | 0 | -4.333 | 4.667 | 5.333 |
| 78-38-6 | 0 | 0 | 0 | 0 | 0 | 0 | 0 |
| 127-31-1 | 0 | 0 | 0.667 | 0 | 0 | 0 | 0 |
| 91-16-7 | 0 | 0 | 0 | 0 | 0 | 0 | 0 |

the NR3C1 gene—which codes for the glucocorticoid receptor—has well-documented associations with metabolic and inflammatory pathways, making it a compelling target for drug repurposing. Similarly, the compounds included in this analysis were chosen for their robust activity scores, reflecting their potential to modulate these targets effectively. This strategic selection ensures that our predictive models focus on biologically relevant relationships, maximizing their translational potential. In summary, out of the 737 enriched genes, 143 genes associated with 6,925 compounds were included in the training set for our model. For each gene target, the number of associated drugs (represented by a value of 1 in the data matrix) ranged from 10 to 223. Conversely, all unassociated drugs with gene targets were marked with a value of 0 in the data matrix. Selected genes are detailed in Table S1.

### Novel gene target prediction

We employed four modeling algorithms by using the Python packages (3.10) of SVC, KNeighborsClassifier, RandomForestClassifier, and XGBClassifier, for the task of gene target prediction. Utilizing compound activity scores as features, our objective was to predict the active or inactive relationship between each gene target and compounds. We developed the k-nearest neighbors (KNN) algorithm, valued for its interpretability. Subsequently, we introduced more sophisticated algorithms, commencing with SVCs, where we explored two different kernels: Radial Basis Function (RBF) and least square. To further augment model performance, tree-based models, namely XGB and Random Forest (RF) were investigated. The selection of four models ensured a comprehensive representation of modeling complexity while embracing popular methodologies in the field. All four modeling algorithms underwent execution on an AWS EC2 instance.

### Fine-tuning and assessment of predictive models.

We performed four different modeling algorithms on all 143 gene targets: 1) KNN; 2) SVC; 3) RF; and 4) XGB. To ensure the robustness and accuracy of the models, we systematically explored various parameter configurations for each modeling algorithm, as detailed in Table S2-S5. If the values of the specific parameters were not specified in our table, default settings were employed. Subsequently, we engaged in hyperparameter fine tuning for all models, utilizing grid-search with 5-fold cross-validation (CV). It operates by exhaustively searching through a specified grid of hyperparameters, systematically evaluating the performance of the model for each combination. The dataset was partitioned into five subsets, with four subsets used for training the model and the remaining subset for validation. This process was repeated five times, with each subset serving as the validation set once. Performance metrics, such as accuracy or mean squared error,

were calculated for each parameter combination across all folds. The optimal parameters were then selected based on the average performance across all folds, providing a robust estimation of model performance while mitigating the risk of overfitting. With the fine-tuned parameters, we assessed model performance with the area under the receiver operating characteristic (ROC) curve (ROC_AUC) score and Area Under the Precision-Recall Curve (AUPRC) in our ML models. The ROC_AUC, ranging from 0 to 1, assesses a model's ability to differentiate between two classes by analyzing the true positive rate (sensitivity) versus the false positive rate (1-specificity) at different decision thresholds. A higher ROC_AUC score reflects better discrimination ability. This score offers a holistic view of model performance across thresholds, making it useful for assessing classification models in diverse fields. In our study, we also calculated the Area Under the Precision-Recall Curve (AUPRC) to address the inherent data imbalance in rare diseases, where positive drug-gene pairs are scarce. While ROC-AUC is a widely used metric, it can yield high values even when predictions favor the majority class, making it less reliable for imbalanced datasets. AUPRC, on the other hand, focuses on the precision and recall of positive cases, excluding true negatives from the calculation. This makes AUPRC a more informative and complementary metric to evaluate machine learning performance in scenarios with significant class imbalance.

## Predictability of Gene Target

The predictability of a given gene target may exhibit variability across distinct modeling frameworks, while the overall predictivity of different genes may vary across these models. To elucidate this phenomenon, we computed the average testing ROC_AUC score for each gene across all models employed in our study. Genes demonstrating consistently higher mean ROC_AUC scores are deemed to possess heightened predictability, whereas those with lower scores are considered less predictable within the scope of the four machine learning models utilized in our analysis

In seeking to explicate the divergent predictability levels among genes, we inquired into whether predictivity correlates with the informational content provided to the model, specifically the number of associated compounds for each gene. Consequently, we calculated Pearson's correlation coefficient between the count of gene-associated compounds and the average ROC_AUC score. Pearson's correlation coefficient, ranging between -1 and 1, quantifies both the strength and directionality of the relationship between two variables. A negative value signifies an inverse correlation, indicating that as one variable changes, the other changes in the opposite direction. Conversely, a positive value denotes a direct correlation, wherein both variables change in tandem. The magnitude of the correlation coefficient reflects the strength of the relationship: higher absolute values indicate a stronger correlation. In our investigation, Pearson's correlation analysis serves to elucidate whether the number of associated compounds is associated with the predictability of genes.

## Parameter Influence on Model Performance

In our pursuit of comprehending the impact of hyper-parameter tuning on the predictivity of the four algorithms, we systematically configured various parameter settings and juxtaposed their average performance across all gene targets. The breadth of parameter settings varies across models, contingent upon the complexity of each algorithm and the extent to which parameters can be feasibly altered. Typically, more intricate models entail a greater number of parameters with a wider range of configurational adjustments. In our exploration, we exhaustively explored parameter spaces, resulting in a myriad of settings across the models under scrutiny. Specifically, we identified 18 configurations for KNN, 54 for SVC, 1621 for RF, and 1154 for XGB. For every model, we categorized these parameter sets into three tiers: those yielding the highest ROC_AUC scores, those with moderate performance, and those resulting in suboptimal outcomes. This approach provided us with a nuanced understanding of the relationship between parameter settings and algorithmic predictivity. Subsequently, we chose 5 parameter sets in each parameter category and visualized the distribution of ROC_AUC scores across all 15 parameter sets for each model using boxplots. By scrutinizing the performance differentials between the best and worst parameter configurations, we aimed to elucidate the extent of parameter influence on algorithmic predictivity. A significant variance in performance between these configurations suggests a robust sensitivity to parameter adjustments,

whereas minimal discrepancies indicate a comparatively stable performance across genes, independent of parameter selection.

## Validation of novel gene-drug pairs with chemical assay

Through fine-tuning the parameter configurations of four distinct algorithms, we identified the top three parameter sets for each algorithm, yielding a total of 12 different models. The selected parameters, along with their corresponding ROC_ AUC scores for the training dataset, were documented in Table 2. Leveraging these models, we predicted the novel gene targets for the Tox 21 chemicals. To accomplish this objective, we computed the disparity between our prediction results and the records from Pharos [53] and the Board Drug Repurposing Hub (BDRH) [54]. If our model predicts a drug-gene relationship with a probability >0.5 but this relationship is not documented in Pharos or BDRH, we consider it as a potential novel gene-drug pair that can be prioritized for experimental validation. Utilizing the top three best parameter configurations for each of the four algorithms, we cross-referenced these 12 distinct prediction results. This analysis yielded a list of candidate gene-drug pairs along with the number of models supporting each prediction. Subsequently, we compared the predictive outcomes from these models with the gene annotations for the Tox21 chemicals obtained from Pharos and the BDRH. Novel gene-drug pairs were identified where the predictive models indicated a connection between a drug and gene target, yet no such association was found in Pharos or BDRH. To validate these novel relationships, we checked the experimental results of compounds with assay data available for their respective targets. For each predicted gene-drug pair, if the compound acted as an active agonist or antagonist for the gene target, our prediction was deemed correct. Conversely, if the compound was inactive against the gene target according to the assay results, our prediction was considered inaccurate. This validation process underscores the robustness of our predictive models.

## Gene-rare disease association identification for drug repurposing application

To achieve the goal of drug repurposing, we aimed to identify possible rare diseases associated with the input chemicals via those newly identified gene targets. We manually searched the OMIM and Orphanet for potential associations

Table 2.  Parameters configuration with top 3 average performance.

| Models | Parameter Configuration 1 | Parameter Configuration 2 | Parameter Configuration 3 |
|---|---|---|---|
| KNN | n_neighbours:33, p:3 | n_neighbours:33, p:2 | n_neighbours:31, p:3 |
| SVM | C:20; gamma:0.5; kernel: rbf | C:15; gamma:0.5; kernel: rbf | C:10; gamma:0.5; kernel: rbf |
| RF | bootstrap: false; max_depth: 6, max_features:auto, min_samples_leaf: 2, min_samples_split:2, n_estimators: 30 | bootstrap: false; max_depth: 6, max_features:auto, min_samples_leaf: 2, min_samples_split:3, n_estimators: 30 | bootstrap: false; max_depth: 6, max_features:sqrt, min_samples_leaf: 2, min_samples_split:2, n_estimators: 30 |
| XGB | Colsample_bytree: 0.6, max_depth: 5, min_child_weight: 3, reg_alpha: 1, subsample: 0.8 | Colsample_bytree: 0.6, max_depth: 3, min_child_weight: 3, reg_alpha: 1, subsample: 0.8 | Colsample_bytree: 0.6, max_depth: 3, min_child_weight: 1, reg_alpha: 1, subsample: 0.6 |

*In KNN model: "n_neighbors" represents the number of neighbors considered when making predictions; "p" represents the power parameter for the Minkowski distance metric. In SVC model: The "C" parameter trades off between achieving a low training error and a low complexity model that generalizes well to unseen data. The "gamma" parameter defines how far the influence of a single training example reaches, with low values meaning 'far' and high values meaning 'close'. In RF model: "Max_depth" specifies the maximum depth of each decision tree in the forest. "Max_features" determines the maximum number of features considered for splitting at each node of a decision tree. "Min_samples_leaf" sets the minimum number of samples required to be at a leaf node. "Min_samples_split" sets the minimum number of samples required to split an internal node. "N_estimators" specifies the number of decision trees to be used in the random forest. In XGB model: "Colsample_bytree" determines the fraction of features (columns) to be randomly sampled for each tree during training. Similar to the parameter in Random Forest, "Max_depth" specifies the maximum depth of each decision tree in the ensemble. "Min_child_weight" determines the minimum sum of instance weight (hessian) needed in a child node. "Reg_alpha" is also known as L1 regularization term. "Reg_alpha" adds penalty to the model for large coefficient values, encouraging sparsity in the feature space. "Subsample" determines the fraction of training data to be randomly sampled for each tree.*

between the gene targets and rare diseases. These results establish connections between drugs, gene targets and rare diseases, thereby advancing the process of drug repurposing for rare conditions.

To further demonstrate our predictive results are useful for supporting drug repurposing, we conducted several case studies. We were able to make connections from chemicals to rare diseases via the predicted gene targets with evidence identified from the NCATS Biomedical Data Translator (Translator) [55]. The linkage found between the compound and disease sheds light on the drug discovery and repurposing.

## Results

In this study, of 7,170 chemical compounds with curve rank data from the Tox21 10K compound library, we collected 6,925 compounds associated with 143 gene targets for novel repurposing target prediction.

### Results on model development

We employed four distinct ML algorithms (SVC, KNN, RF, XGB) with the parameter set that yielded the best average performance. The optimized parameters are listed in Table 3. All four machine learning algorithms demonstrated strong performance, with ROC-AUC values exceeding 0.75. Our results show that the SVC, XGB, and RF models achieved high AUPRC values, indicating their robustness in handling imbalanced data. In contrast, the KNN model exhibited a tendency to favor true negatives, leading to comparatively lower AUPRC scores. This behavior is likely due to the simplicity of the KNN algorithm, which relies on identifying the closest data points in the training set and involves minimal model building and hyperparameter tuning. The overall performance of KNN, SVC, RF, and XGB on the training and test sets is depicted respectively in Fig 1, ROC_AUC scores were utilized to assess model performance as the y-axis in Fig 1. Notably, KNN exhibited the lowest performance compared with the other three algorithms with lowest AUPRC and ROC_AUC, while SVC emerged as the top performer. It's worth mentioning that the predictivity variation across all gene targets was minimal in SVC as well, whereas RF and XGB displayed larger varying degrees of performance.

We evaluated the model performance regarding overfitting and underfitting, as showcased in Fig 1. Impressively, all four algorithms demonstrated performance exceeding 0.7 on the test dataset. Moreover, RF and XGB exhibited enhanced performance on the test data, surpassing 0.8, while SVC and KNN displayed similar performance trends as the training dataset. These results underscore the better predictivity of tree-based models for gene targets compared to other algorithms.

In summary, our prediction models trained on the Tox21 dataset displayed robust performance across all four algorithms with ROC_AUC scores exceeding 0.7. Notably, XGB emerges as the standout performer, showing the best performance on both the training and test datasets, solidifying its status as the premier prediction algorithm among the four algorithms utilized in this study.

### 2. Predictability of Gene Targets

Apparently, the predictability of gene targets exhibits variations among genes based on the predictive results. It is acknowledged that the ability to accurately forecast the interactions between chemical compounds and specific genes is

**Table 3. Selected Best Parameters.**

| Models | Parameters | Mean ROC_AUC [*] | Mean AUPRC [*] |
|--------|-----------|------------------|----------------|
| KNN | n_neighbours:33, p:3 | 0.71268 | 0.50421 |
| SVC | c:20, gamma:0.5, kernel: rbf | 0.77428 | 0.80494 |
| RF | bootstrap: false; max_depth: 6, max_features:auto, min_samples_leaf: 2, min_samples_split:2, n_estimators: 30 | 0.75529 | 0.73537 |
| XGB | Colsample_bytree: 0.6, max_depth: 5, min_child_weight: 3, reg_alpha: 1, subsample: 0.8 | 0.77504 | 0.73602 |

[*]*The mean ROC_AUC/AUPRC was calculated by averaging the ROC_AUC/AUPRC scores across all 143 selected gene targets.*

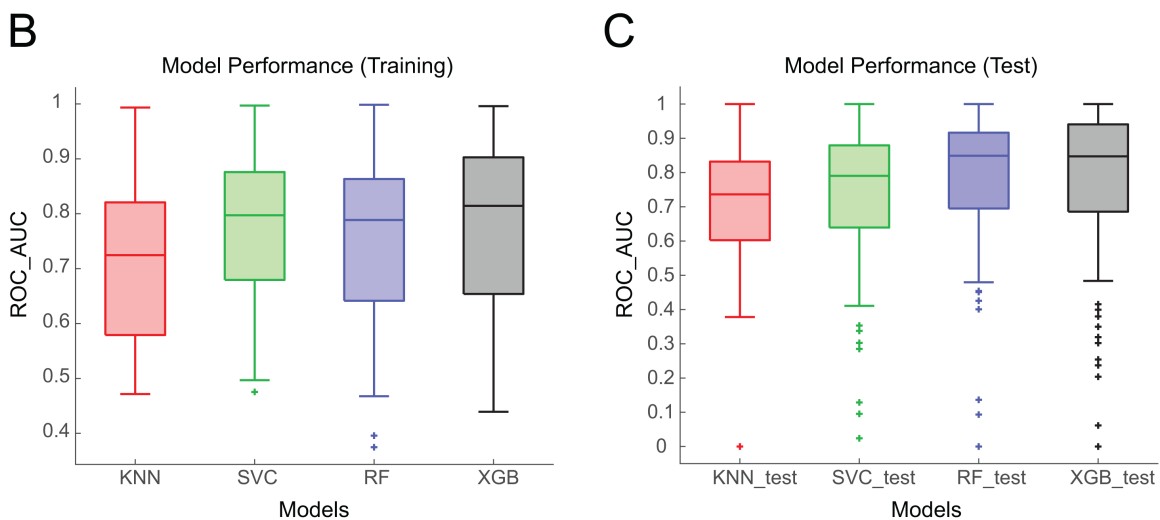

A    Table_Selected Best Parameters

| Model | Parameter | Mean ROC_AUC Score |
|---|---|---|
| KNN | n_neighbors: 33, p: 3 | 0.7126818046805555 |
| SVC | C: 20, gamma: 0.5, kernel: rbf | 0.7742757875524475 |
| RF | bootstrap: False, max_depth: 6, max_features: auto, min_samples_leaf: 2, min_samples_split: 2, n_estimators: 30 | 0.7552852895998671 |
| XGB | colsample_bytree: 0.6, max_depth: 5, min_child_weight: 3, reg_alpha: 1, subsample: 0.8 | 0.7750385602440503 |

**Fig 1. General model performance for different algorithms. A)** Model performance on the training set. **B)** Model performance on the test set. For all plots, "+ " denotes the outliers that fall significantly outside the range of the other data points in the dataset.

influenced by a multitude of factors, such as spanning data quality [56, 57] computational methodologies [27], and Biological Context [58]. Thus, we next aimed to delve deeper into the predictability of the 143 gene targets central to our study.

To assess the predictability of gene targets, we generated a heat map (Fig 2A) with the ROC_AUC calculated for all 143 gene targets across four different ML algorithms. As shown in Fig 2A, each cell in this heat map is the average ROC_AUC score based on the cross-validation results. The map contains 143 rows corresponding to 143 gene targets, and the 4 columns for KNN, RF, SVC, and XGB algorithms, respectively. Notably, certain genes, exemplified by NR3C1, SERPINA6, and PGR, consistently demonstrated high ROC_AUC values regardless of the modeling algorithm employed. Conversely, several genes are consistently associated with low ROC_AUC values across all models. Obviously XGB and SVC showed better performance among most of the gene targets, which is shown in red in Fig 2A.

Further elucidating the predictability landscape, Fig 2B shows the top 20 genes with high predictability across all four predictions. Particularly, the TUBB gene emerged with the highest predictability in both models of SVC and RF. Genes such as NR3C1, NR3C2, HTR2A, HTR2C, and KCNJ6 remain high predictability across all models. Meanwhile, we observed that genes illustrate different performance variability across 4 models. For example, although NR3C1 performs the third top ranked gene with the highest predictability across all models, KNN, and SVC predict the genes with less variability across different parameters. However, the predictability of NR3C1 depends more on hyperparameter tuning with XGB models.

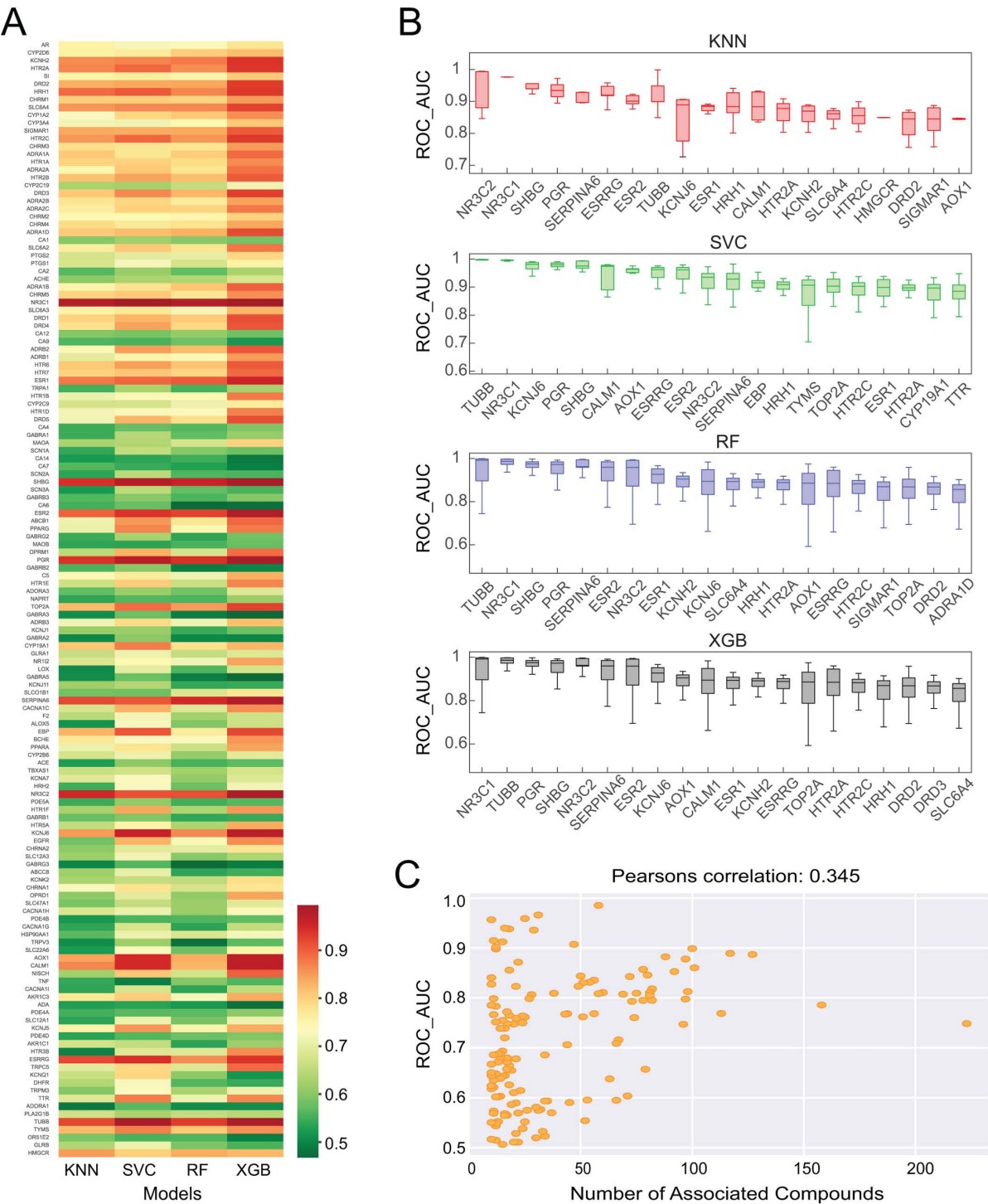

**Fig 2. Gene targets prediction assessment. A)** Heatmap of ROC_AUC scores across different modeling algorithms and gene targets. High ROC_AUC scores are highlighted in red, whereas low ROC_AUC scores in green. **B)** The boxplot illustrates the top 20 gene targets with the highest ROC_AUC scores for KNN, SVC, RF, and XGB. Within each boxplot, the gene targets are ordered based on their median ROC_AUC scores, from highest to lowest. **C)** The correlation between ROC_AUC scores and the number of associated compounds that each gene targets. Each yellow dot denotes one gene target.

In our final analysis, we identified 16 gene targets consistently exhibiting low ROC_AUC values (<0.6) across all four algorithms, as outlined in Table S6. To probe the underlying factors contributing to the divergent predictability among these genes, we hypothesized that the number of associated compounds might significantly impact the predictive results. By employing Pearson's correlation coefficient, we found weak positive correlations between the number of associated compounds and ROC_AUC scores, with Pearson's correlation coefficient registering at 0.345 (Fig 2C). Despite this correlation, the weak magnitude prompts consideration of alternative factors causing the variability in gene predictivity, such as the diversity of compounds associated with a gene target.

In summary, our models effectively predicted the remaining 127 gene targets, encompassing nearly 90% of the total gene targets examined. This noteworthy achievement underscores the reliability and robustness of our predictive framework, thereby validating its potential to expedite accurate gene target prediction within the realms of drug discovery and molecular biology research.

## Parameter influence on model performance

In our quest to comprehend the extent to which hyper-parameter tuning affects the prediction, we configured different parameter settings and compared their average performance across all gene targets. Fifteen sets of parameters for each model were selected based on their performance, including the top five performers, the bottom five performers, and those in the middle tier of performance (Listed in Table S7). Illustrated in Fig 3, the median performance of various configurations for KNN, RF, SVC, and XGB unveils intriguing insights.

Our analysis reveals that parameter fine-tuning is highly impacting on the RF model, while the performance of the KNN algorithm exhibits relatively minimal fluctuations with varying hyper-parameters. Notably, the RF model consistently yields ROC_AUC values exceeding 0.8 with a particular set of hyper parameters. However, the XGB model demonstrated persistent performance with an average ROC_AUC of around 0.75 regardless of the hyper-parameters. Collectively, our findings disclosed the profound impact of hyper-parameter selection on certain modeling algorithms, such as RF, while exerting less influence on others, such as KNN, and XGB has higher stability across all configurations.

## Results on validation of candidate gene-drug pairs

The primary objective of constructing prediction models is to identify potential new gene-drug connections for drug repurposing. To validate our predictions, we explored available in vitro assay data to see if any of the predicted gene-drug pairs have experimental support.

The process of identifying potentially new gene-drug connections is described in Methods. We compared our predictions with records from Pharos and the Board Drug Repurposing Hub (BDRH). If our model predicts a connection with probability >0.5 that was not documented in Pharos or BDRH, we identify it as a potential novel gene-drug pair for further experimental validation. We utilized the top three parameter configurations for each of the four algorithms, resulting in 12 distinct predictions. Cross-referencing these predictions produced a list of candidate gene-drug pairs along with the count of models supporting each prediction. In total, we uncovered 220 gene-drug pairs supported by at least one ML model but not documented in Pharos or the BDRH (Examples are shown in Table 4 and the full list can be found in Table S8). We manually identified 60 gene-drug pairs that have in vitro assay data available, 52 of which were supported by experimental results, that is, the compounds acted either as active agonists or active antagonists of their respective targets in these assays, implying a confirmation rate (86.7%) exceeding 85% (Examples are shown in Table 5 and full list in Table S9).

## Results on identifying gene-rare disease associations for drug repurposing

We conducted a manual search of the OMIM and Orphanet database to ascertain whether the gene targets identified in our novel gene-drug pairs are associated with any rare diseases. The findings are summarized in Table 6, revealing a total

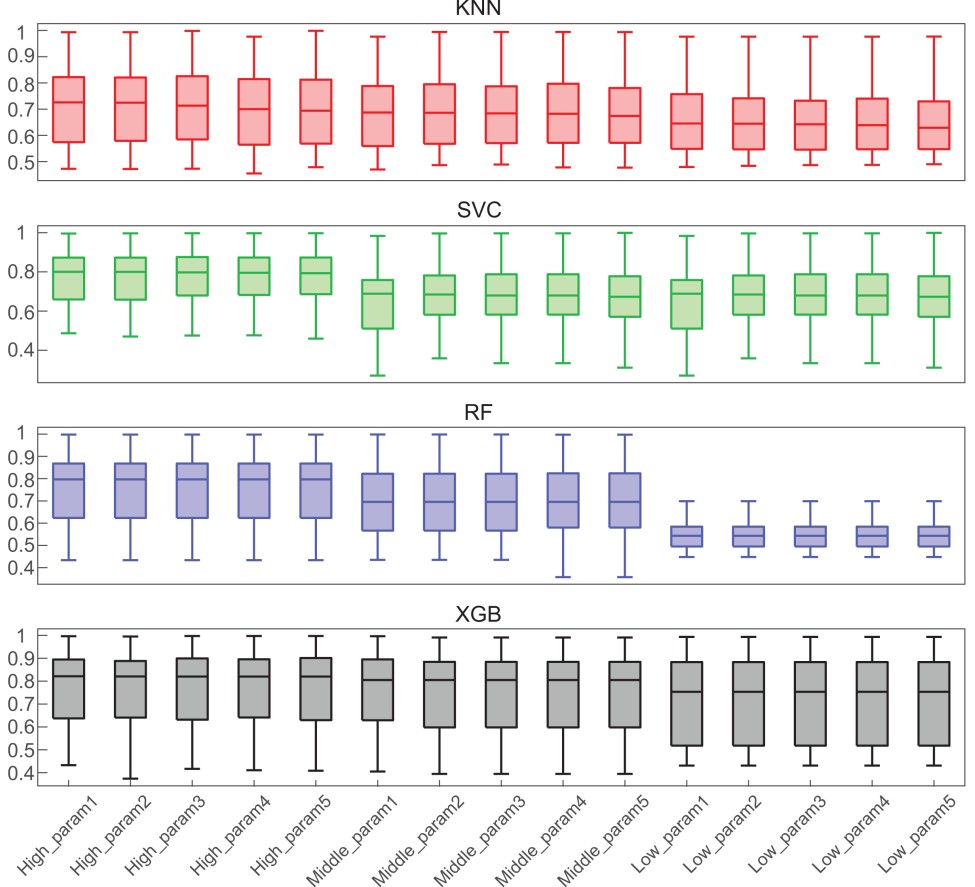

**Fig 3. Impact of hyper-parameter fine-tuning on model performance.** The boxplot showcases the 15 ROC_AUC scores, comprising the 5 highest, 5 intermediate, and 5 lowest, for KNN, SVC, RF, and XGB. For each boxplot, the parameter configurations are ordered based on their median ROC_AUC scores, from highest to lowest.

**Table 4. Examples of gene-drug pairs predicted by different models.**

| Models | Interest Gene Targets | Structure ID |
|---|---|---|
| [KNN_1, KNN_2, KNN_3, RF_1, RF_2, RF_3, SVC_1, SVC_2, SVC_3, XGB_1, XGB_2, XGB_3] | 'NR3C1' | 127-31-1 |
| [KNN_1, KNN_2, KNN_3, RF_1, RF_2, RF_3, SVC_1, SVC_2, SVC_3, XGB_1, XGB_2, XGB_3] | 'NR3C1' | 1310709-74-0 |
| [KNN_1, KNN_2, KNN_3, RF_1, RF_2, RF_3, SVC_1, SVC_2, SVC_3, XGB_1, XGB_2, XGB_3] | 'NR3C1' | 59198-70-8 |
| [KNN_2, RF_1, RF_2, RF_3, SVC_1, SVC_2, SVC_3, XGB_1, XGB_2, XGB_3] | 'SHBG' | 10161-33-8 |
| [SVC_1, SVC_2, SVC_3, XGB_1, XGB_2, XGB_3] | 'ACHE' | 102518-79-6 |
| [RF_1, RF_3, SVC_3, XGB_1, XGB_2, XGB_3] | 'ADRB3' | 32266-10-7 |
| [SVC_1, SVC_2, SVC_3, XGB_1, XGB_2, XGB_3] | 'DRD4' | 75444-65-4 |
| [KNN_1, KNN_3, RF_1, RF_3, XGB_1, XGB_2] | 'HTR2A' | 75859-03-9 |
| [KNN_1, KNN_3, RF_2, XGB_1, XGB_2, XGB_3] | 'HTR2C' | 75859-03-9 |

of 35 distinct rare diseases linked to various gene targets for which potential drugs have been identified. Further exploration of these disease-drug relationships promises to provide insights into drug repurposing strategies tailored for rare diseases.

**Table 5. Examples of gene-drug pairs validated by Tox21 in vitro assay data.**

| Gene Targets | Structure ID | Assay Names | Assay Outcome |
|---|---|---|---|
| 'CYP1A2' | 81-14-1 | tox21-p450-1a2-p1 | active antagonist |
| 'CYP1A2' | 15930-66-2 | tox21-p450-1a2-p1 | active antagonist |
| 'CYP2D6' | 303-49-1 | tox21-p450-2d6-p1 | active antagonist |
| 'CYP2D6' | 1104-22-9 | tox21-p450-2d6-p1 | active antagonist |
| 'CYP2D6' | 113-92-8 | tox21-p450-2d6-p1 | active antagonist |
| 'DRD2' | 959-24-0 | tox21-drd2-agonist-p1/tox21-drd2-antagonist-p1 | inactive/inactive |
| 'ESR1' | 26538-44-3 | tox21-er-luc-bg1-4e2-antagonist-p1 | active antagonist |
| 'KCNH2' | 75859-03-9 | tox21-herg-u2os-p1 | active antagonist |
| 'KCNH2' | 553-08-2 | tox21-herg-u2os-p1 | active antagonist |
| 'NR3C1' | 127-31-1 | tox21-gr-hela-bla-antagonist-p1/tox21-gr-hela-bla-agonist-p1 | active agonist |

## Application of novel gene targets for drug repurposing

The principal objective of this study is to advance the frontier of drug repurposing. Consequently, the elucidation of a cohesive relationship between pharmaceutical compounds and pathological conditions assumes paramount importance. We are now embarking on a quest to identify rare diseases intricately linked to specific gene targets by utilizing the OMIM and Orphanet databases. Our aim in this phase is to outline various pathways that elucidate how drugs implicated in the modulation of these genes may offer therapeutic avenues for the treatment of the rare diseases. Through the following examples, we demonstrate how our research findings facilitate the establishment of robust connections between rare diseases and pharmaceutical agents via the identified novel gene targets, thereby presenting promising avenues for therapeutic exploration and intervention.

**Case study 1. Candidate Drugs for Generalized Glucocorticoid Resistance (GCCR).** The NR3C1 gene has been predicted as a novel gene for eight distinct compounds, as listed in Table 7. GCCR (GARD:0002499) is a rare adrenogenital syndrome characterized by generalized, partial tissue insensitivity to glucocorticoids, and it is caused by heterozygous mutation in the glucocorticoid receptor gene (NR3C1) on chromosome 5q31 [59]. With the connections between eight chemical compounds and the gene NR3C1, and NR3C1 and GCCR, we aimed to explore the potential use of these drugs in treating GCCR.

Through our analysis of scientific evidence mined from the Translator ecosystem, we found that Fludrocortisone, Rimexolone, and Fluoxymesterone can modulate the NR3C1 gene mainly by binding to it and activating its signaling pathway [60–62] and subsequently influencing GCCR by restoring the function of the glucocorticoid receptor [63–66]. These three drugs as synthetic corticosteroids are commonly used to treat adrenal insufficiency diseases like Addison's disease [62,67–69]. Additionally, research indicates that GCCR patients typically exhibit deficiencies in adrenal corticosteroids, including cortisol and aldosterone [70–73]. Therefore, Fludrocortisone, Rimexolone, and Fluoxymesterone may potentially treat GCCR by providing adrenal corticosteroids. Rimexolone is shown as an example in Fig 4, other examples can be found under the column 'Scientific evidence from the Translator' in Table 7.

Moreover, Rimexolone and Flunisolide, two of the aforementioned drugs, are known to affect the glucocorticoid receptor (GR) by binding to it with high affinity [74]. As cortisol action mediated by the GR is diminished in GCCR patients [75, 76], Rimexolone and Flunisolide may also be effective in treating GCCR by enhancing GR binding. Hormone replacement therapy is another pivotal approach to maintaining hormonal balance in patients [77,78], which could aid in treating GCCR. Among the eight drugs, Diflucortolone valerate and Melengestrol acetate are connected to GCCR through hormone pathways, suggesting their potential use in hormone replacement therapy. Additionally, Deoxycorticosterone acetate (DOCA) is documented as a drug that induces one of the GCCR phenotypes, hypertensive disorder. Research shows that DOCA is often used to induce hypertension in animal models [79–81], thus reducing DOCA levels may partially alleviate GCCR symptoms.

**Table 6. Target Gene and Related Rare Diseases based on OMIM and Orphanet.**

| Gene Targets | Rare Disease Records | OMIM Number |
|---|---|---|
| ACHE | YT BLOOD GROUP ANTIGEN | OMIM: 112100 |
| ADRA2A | LIPODYSTROPHY, FAMILIAL PARTIAL, TYPE 8; FPLD8 | OMIM: 620679 |
| ADRB1 | RESTING HEART RATE, VARIATION IN | OMIM: 607276 |
| ADRB1 | SHORT SLEEP, FAMILIAL NATURAL, 2; FNSS2 | OMIM: 618591 |
| AR | Spinal and bulbar muscular atrophy, X-linked 1 | OMIM: 313700 |
| AR | Hypospadias 1, X-linked | OMIM: 30633 |
| AR | ANDROGEN INSENSITIVITY, PARTIAL; PAIS | OMIM: 312300 |
| AR | ANDROGEN INSENSITIVITY SYNDROME; AIS | OMIM: 300068 |
| BCHE | BUTYRYLCHOLINESTERASE DEFICIENCY; BCHED | OMIM: 617936 |
| C5 | ECULIZUMAB, POOR RESPONSE TO | OMIM: 615749 |
| C5 | COMPLEMENT COMPONENT 5 DEFICIENCY; C5D | OMIM: 609536 |
| CA12 | Hyperchlorhidrosis, isolated | OMIM: 143860 |
| CA2 | Osteopetrosis, autosomal recessive 3, with renal tubular acidosis | OMIM: 259730 |
| CALM1 | Long QT syndrome 14 | OMIM: 616247 |
| CALM1 | Ventricular tachycardia, catecholaminergic polymorphic, 4 | OMIM: 614916 |
| CHRM3 | PRUNE BELLY SYNDROME; PBS | OMIM:100100 |
| CYP19A1 | AROMATASE DEFICIENCY | OMIM: 613546 |
| CYP19A1 | Aromatase excess syndrome | OMIM: 139300 |
| CYP2C9 | COUMARIN RESISTANCE | OMIM: 12270 |
| DRD3 | TREMOR, HEREDITARY ESSENTIAL, 1; ETM1 | OMIM: 190300 |
| ESR1 | BREAST CANCER, FAMILIAL MALE, INCLUDED | OMIM: 114480 |
| GABRB2 | Developmental and epileptic encephalopathy 92 | OMIM: 617829 |
| GABRG2 | Generalized epilepsy with febrile seizures plus, type 3 | OMIM: 607681 |
| GABRG2 | FEBRILE SEIZURES, FAMILIAL, 8; FEB8 | OMIM: 607681 |
| GABRG2 | DEVELOPMENTAL AND EPILEPTIC ENCEPHALOPATHY 74; DEE74 | OMIM: 618396 |
| KCNH2 | LONG QT SYNDROME 2; LQT2 | OMIM: 613688 |
| KCNH2 | SHORT QT SYNDROME 1; SQT1 | OMIM: 609620 |
| NR3C1 | GLUCOCORTICOID RESISTANCE, GENERALIZED; GCCR | OMIM: 615962 |
| PPARG | LIPODYSTROPHY, FAMILIAL PARTIAL, TYPE 3; FPLD3 | OMIM: 604367 |
| SI | SUCRASE-ISOMALTASE DEFICIENCY, CONGENITAL; CSID | OMIM: 609845 |
| SIGMAR1 | AMYOTROPHIC LATERAL SCLEROSIS 16, JUVENILE; ALS16 | OMIM: 614373 |
| SIGMAR1 | NEURONOPATHY, DISTAL HEREDITARY MOTOR, AUTOSOMAL RECESSIVE 2; HMNR2 | OMIM: 605726 |
| SLC6A2 | ORTHOSTATIC INTOLERANCE | OMIM: 604715 |
| SLC6A3 | PARKINSONISM-DYSTONIA 1, INFANTILE-ONSET; PKDYS1 | OMIM: 613135 |
| SLCO1B1 | HYPERBILIRUBINEMIA, ROTOR TYPE; HBLRR | OMIM: 237450 |

In summary, our study identifies several drugs with the potential to treat GCCR through different pathways, including increasing adrenal corticosteroids, enhancing GR binding efficiency, hormone replacement therapy and alleviating GCCR symptoms. Meanwhile, there are other compounds, e.g., Cortodoxone, that can be further investigated and shows weak relationship to GCCR according to current research recorded in Translator.

**Case study 2. Candidate Drugs for short QT syndrome.** In our results, KCNH2 was predicted as a novel gene target of eleven drugs listed in Table 8. KCNH2 provides instructions for making channels that transport positively charged atoms (ions) of potassium out of cells, and mutations in the KCNH2 gene can cause short QT syndrome (OMIM: 609620) [82]. Thus, we speculated that drugs linked to KCNH2 might present promising candidates for short QT syndrome treatment.

**Table 7. Candidate compounds for GCCR.**

| Compounds | Original indication | Use for GCCR | Scientific evidence from the Translator |
|---|---|---|---|
| Rimexolone | Employed in ophthalmology to treat eye inflammation and allergic eye diseases like keratitis and conjunctivitis | Providing adrenal corticosteroids; Enhancing GR binding | https://arax.ncats.io/?r=236272 |
| Fluoxymesterone | Treatment of hypogonadism | Providing adrenal corticosteroids; Enhancing GR binding | https://arax.ncats.io/?r=241052 |
| Fludrocortisone | Adrenal insufficiency diseases like Addison's disease | Providing adrenal corticosteroids | https://arax.ncats.io/?r=236165 |
| Melengestrol acetate | Growth-promoting agent in livestock | Hormone replacement therapy | https://arax.ncats.io/?r=241051 |
| Deoxycorticosterone acetate | Adrenal insufficiency diseases | Alleviate one GCCR symptom, hypertension | https://arax.ncats.io/?r=236273 |
| Diflucortolone valerate | Combat skin inflammation and allergic reactions such as eczema and dermatitis | Hormone replacement therapy | https://arax.ncats.io/?r=241047 |
| Cortodoxone | Adrenal insufficiency diseases like Addison's disease | Weak relationship through ethanol. | https://arax.ncats.io/?r=236270 |
| Halometasone hydrate | Combat skin inflammation and allergic reactions such as eczema and dermatitis | N/A | https://arax.ncats.io/?r=241048 |

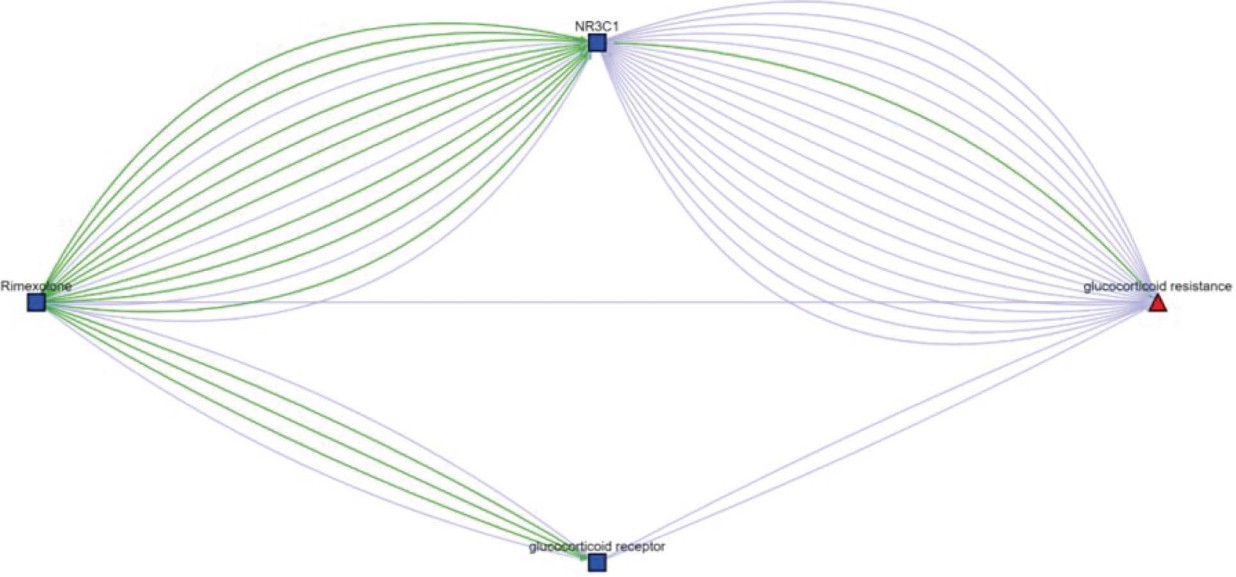

**Fig 4. Associations between Rimexolone and Generalized Glucocorticoid Resistance from the Translator.** (the original graph can be accessed via https://arax.ncats.io/?r=236272).

To test our hypothesis, we examined the scientific evidence identified from the Translator, and among these eleven drugs, we found that Bucindolol, triprolidine, Cyproheptadine Hydrochloride, and Thonzonium bromide are connected to short QT syndrome via the KCNH2 gene. Thonzonium bromide is shown as an example in Fig 5, The KCNH2 gene, also referred to as the hERG gene, encodes a protein forming channels in cardiac muscle cell membranes [83]. These channels regulate potassium ion flow, crucial for cardiac rhythm maintenance [84,85]. Cyproheptadine has been demonstrated to interfere with the hERG channel function, while Bucindolol and triprolidine reduce the KCNH2 protein activity [86]. In

**Table 8. Candidate compounds for short QT syndrome.**

| Compounds | Original indication | Use for short QT syndrome | Scientific evidence from the Translator |
|---|---|---|---|
| Bucindolol | Management of heart failure and hypertension | KCNH2 deactivation; β-blocker of calcium channel | https://arax.ncats.io/?r=241069 |
| Thonzonium bromide | Used in topical formulations for its ability to inhibit the growth of bacteria, fungi, and other microorganisms | KCNH2 deactivation | https://arax.ncats.io/?r=241079 |
| triprolidine | Symptomatic relief of allergic conditions such as hay fever | KCNH2 deactivation | https://arax.ncats.io/?r=241074 |
| Cyproheptadine hydrochloride | Allergic conditions; Serotonin Syndrome; Migraine Prophylaxis | KCNH2 deactivation | https://arax.ncats.io/?r=241077 |
| iloperidone | Treatment of schizophrenia | Induced tachycardia | https://arax.ncats.io/?r=241071 |
| Clomipramine hydrochloride | Treatment of various mental health conditions, including: Obsessive-Compulsive Disorder (OCD), panic disorder. | Weak relationship through Chloride ion | https://arax.ncats.io/?r=241073 |
| Trimipramine maleate | Treatment of depression; anxiety disorders; insomnia | Weak relationship connected by prelamin-A/C | https://arax.ncats.io/?r=241078 |
| Mebeverine hydrochloride | Alleviate symptoms associated with irritable bowel syndrome (IBS) and related gastrointestinal disorders | N/A | https://arax.ncats.io/?r=241070 |
| Rimcazole dihydrochloride | Preclinical studies only | N/A | https://arax.ncats.io/?r=241072 |
| Promethazine | Allergic conditions; nausea and vomiting; sedation and anxiolysis; insomnia | N/A | https://arax.ncats.io/?r=241075 |
| AVE8923 | potential therapeutic agent for cardiovascular conditions | N/A | Cannot find this drug in translator |

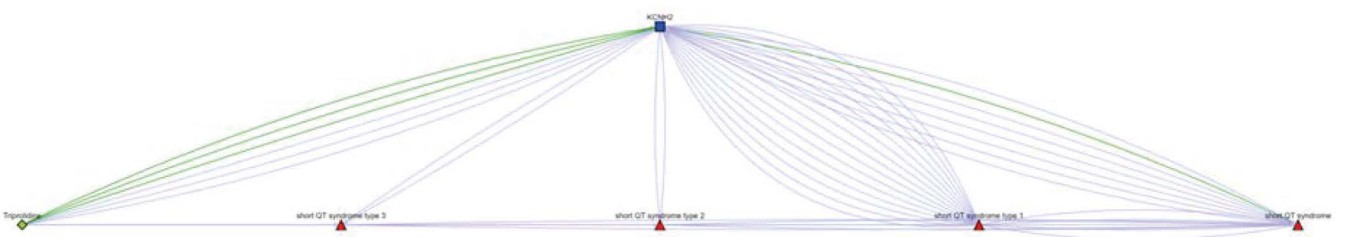

**Fig 5. Associations between triprolidine and short QT syndrome from the Translator.** (the original graph can be accessed via https://arax.ncats.io/?r=241074).

the meanwhile, research shows that accelerated KCNH2 deactivation is linked to arrhythmogenesis, a key symptom of short QT syndromes [87]. Thus, these drugs hold potential in short QT syndrome treatment through KCNH2 deactivation. Notably, Bucindolol, a β-blocker formerly employed in heart-related conditions, has been phased out due to its ambiguous efficacy [88,89].

Although no published studies on iloperidone specifically address its relationship with KCNH2 gene targets, we identified the potential indication of iloperidone for short QT syndrome via Translator. Notably, iloperidone-induced tachycardia, a short QT syndrome phenotype, suggests its relevance (table reference column). Hence, iloperidone warrants investigation as a potential treatment for cardiac diseases with similar phenotypes.

In conclusion, our findings suggest that drugs identified in our study can modulate KCNH2 gene target function, holding promise for short QT syndrome treatment.

## Discussion

In our study, we introduced a systematic approach to identify novel relationships between chemical compounds and gene targets and rare diseases based on biological activity profiles generated by using Tox21 bioassay screening data towards drug repurposing. With the developed ML models achieving an overall prediction accuracy reflected by ROC-AUC scores exceeding 0.7 and showing comparable values in APRUC metrics, we were able to identify 220 potential pairs of gene targets and compounds. Of these, 60 pairs had public assay records, with 52 of them validated by experimental outcomes. In addition, the two case studies further underscore the robustness of our approach in predicting novel gene targets for drug repurposing and disease treatment initiatives.

Research indicates that algorithms with more intricate architectures, such as SVC or tree-based ML models, often exhibit better statistical performance. Our study corroborates this trend, as our results consistently demonstrate that RF, SVC, and XGB consistently outperform KNN models across all 143 genes analyzed. Furthermore, our investigation into hyper-parameter tuning revealed an interesting pattern: while fine-tuning had a significant impact on RF algorithms, its effect was less pronounced in KNN and XGB models. This phenomenon can be attributed to the intricate nature of the models and the multitude of hyper-parameters that can be fine-tuned to optimize performance. Thus, the complexity of the algorithm architecture and the flexibility in parameter adjustments likely contribute to the varying degrees of sensitivity to fine-tuning observed across different ML models.

In our study, we have identified 16 gene targets that exhibit a lack of predictability. The predictability of gene targets, defined as the capacity of ML models to accurately predict interactions between chemical compounds and specific genes, is influenced not only by computational algorithms but also by various other factors such as data quality, data quantity, and biological complexities. To delve deeper into this issue, we have explored the correlation between the quantity of associated compounds and the predictability of gene targets. Our analysis reveals a slightly positive correlation between the number of known compounds of gene targets and their predictability. It appears that gene targets with a limited number of associated compounds tend to exhibit poorer model performance. This could be the first reason that hinders the predictability of these 16 genes (maximum number of related compounds: 52; minimal number of related compounds: 10; Mean: 22). Conversely, genes involved in well-characterized cellular pathways or disease processes may demonstrate higher predictability due to their well-understood functions and interactions. Additionally, the lack of associated compounds further indicates a poor understanding of these gene targets, contributing to their low predictability across different models. Thus, it becomes evident that the absence of sufficient biological context significantly impacts the prediction accuracy of these gene targets.

To accomplish our primary objective of drug repurposing using our ML models, we have successfully identified 220 gene-target pairs for drugs that were not previous reported. We checked 60 pairs that have in vitro assay data and found that 52 pairs were confirmed by experimental evidence. We also found that NR3C1 genes are associated with multiple compounds that are supported by nearly all different predictive models. The NR3C1 gene codes for the glucocorticoid receptor (GR). Changes in NR3C1 gene targets lead to not only common diseases like polycystic ovarian syndrome but also result in rare diseases like GCCR. Besides that, we also found a great number of compounds related to the family of HTR2, PTGS, and DRD gene targets. All these gene targets are highly related to common and rare diseases such as schizophrenia, gastric ulcer, urticaria, and endogenous depression. Thus, predicting the potential drugs associated with these gene targets will provide meaningful information for drug repurposing and future clinical studies.

Besides those gene-drug pairs that have been validated with Tox21 assay data, the newly predicted drug candidates from this study provide additional opportunities for future investigation. For example, our model predicts the novel association between metoclopramide and the gene ADRA2A. ADRA2A is related to Familial Partial Lipodystrophy (FPLD) [90], and metoclopramide is connected to different kinds of FPLD through multiple pathways including hypertensive disorder [91]. Thus, we hypothesized this might be a new potential solution for treating FPLD with metoclopramide via ADRA2A.

In general, the findings of this study hold significant translational potential for addressing the unmet medical needs in rare diseases. By utilizing machine learning models, we identified novel gene-drug associations that provide a foundation for targeted therapeutic development. For instance, the high predictability of gene targets such as NR3C1 and KCNH2, coupled with their relevance to rare diseases like Generalized Glucocorticoid Resistance and Short QT Syndrome, demonstrates the potential to prioritize drug candidates for preclinical and clinical evaluations. Furthermore, the diversity of associated compounds for these targets enables the exploration of multiple therapeutic pathways. This approach is particularly valuable for rare diseases, where limited research funding often constrains the development of targeted treatments. By focusing on well-characterized gene-disease relationships, such as those supported by existing evidence, this study makes it feasible to design clinical studies with higher confidence in the underlying biology. Our methodology also has broader implications for drug repurposing in rare diseases. The ability to predict novel gene-drug relationships and validate them experimentally highlights a systematic pipeline that can be applied to other underexplored diseases. This not only accelerates the identification of promising therapeutic candidates but also provides a scalable framework for addressing the broader challenges of drug discovery in rare disease contexts.

Despite the promising results from our models, our study acknowledges several limitations. The imbalance of the dataset, characterized by a disproportionately low number of positive drug-gene pairs, can lead to skewed predictions favoring the major class. While we employed metrics like AUPRC to mitigate this, future studies would benefit from augmented datasets with more balanced distributions. Additionally, the varying performance of machine learning models highlights the need to balance model complexity and interpretability. Simpler models like KNN underperform due to their inability to capture intricate relationships, while more complex models like XGB, although more accurate, present challenges in biological interpretation due to their computational complexity. Finally, the predictability of certain gene targets is constrained by limited biological context, emphasizing the importance of integrating additional functional data into future analyses.

## Supporting information

**Table S1. Selected gene list.**
(CSV)

**Table S2. Knn parameterSets.**
(XLSX)

**Table S3. SVC parameterSets.**
(XLSX)

**Table S4. RF parameterSets.**
(XLSX)

**Table S5. XGB parameterSets.**
(XLSX)

**Table S6. Gene with low predictability.**
(CSV)

**Table S7. High middle low performance parameters.**
(XLSX)

**Table S8. All gene compound pairs.**
(XLSX)

**Table S9. Gene compound pairs validated byTox21.**
(XLSX)

## Acknowledgements

We would like to thank Dr. Shixue Sun for his helpful discussion on study design. The analyses described in this publication were conducted with data and/or tools accessed through the NCATS Biomedical Data Translator. ([https://ncats.nih.gov/translator](https://ncats.nih.gov/translator)).

## Author contributions

**Conceptualization:** Qian Zhu.

**Data curation:** Binghan Xue.

**Formal analysis:** Binghan Xue.

**Investigation:** Binghan Xue.

**Methodology:** Binghan Xue.

**Project administration:** Qian Zhu.

**Resources:** Yanji Xu, Ruili Huang.

**Supervision:** Qian Zhu.

**Validation:** Binghan Xue.

**Visualization:** Binghan Xue.

**Writing – original draft:** Binghan Xue.

**Writing – review & editing:** Binghan Xue, Ruili Huang, Qian Zhu.
**Funding**

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
