## [Decision Letter · Decision Letter 0]

10 Dec 2024

PONE-D-24-43717Novel Target Identification towards Drug Repurposing Based on Biological Activity ProfilesPLOS ONE

Dear Dr. Zhu,

Thank you for submitting your manuscript to PLOS ONE. After careful consideration, we feel that it has merit but does not fully meet PLOS ONE’s publication criteria as it currently stands. Therefore, we invite you to submit a revised version of the manuscript that addresses the points raised during the review process.

We look forward to receiving your revised manuscript.

Kind regards,

Sheikh Arslan Sehgal, PhD

Academic Editor

PLOS ONE

**Journal Requirements:**

We would like to thank Dr. Shixue Sun for his helpful discussion on study design and reading drafts of this manuscript. This project was supported by the intramural programs (Grant number: ZIC TR000410-05) at the National Center for Advancing Translational Science (NCATS). The analyses described in this publication were conducted with data and/or tools accessed through the NCATS Biomedical Data Translator. (https://ncats.nih.gov/translator). 

"N/A"

"N/A"

Reviewers' comments:

Reviewer's Responses to Questions

**Comments to the Author**

1. Is the manuscript technically sound, and do the data support the conclusions?

Reviewer #1: Yes

Reviewer #2: Yes

2. Has the statistical analysis been performed appropriately and rigorously? 

Reviewer #1: Yes

Reviewer #2: Yes

3. Have the authors made all data underlying the findings in their manuscript fully available?

Reviewer #1: Yes

Reviewer #2: Yes

4. Is the manuscript presented in an intelligible fashion and written in standard English?

Reviewer #1: Yes

Reviewer #2: Yes

5. Review Comments to the Author

**Reviewer #1: ** The manuscript "Novel Target Identification towards Drug Repurposing Based on Biological Activity Profiles"

addresses an important and timely topic in the field of drug discovery, focusing on drug repurposing through advanced machine learning models and the integration of biological activity profiles. The authors provide a comprehensive approach to target identification, utilizing robust computational methodologies and validating their predictions experimentally.

The study's strengths include:

(i) Clear Objective: The focus on rare diseases and the application of drug repurposing to address their unmet therapeutic needs is highly relevant. (ii) Methodological Rigor: The use of multiple machine learning algorithms (e.g., SVC, RF, XGB) and thorough parameter optimization enhances the robustness of the findings. (iii) Validation: Experimental confirmation of the predicted gene-drug pairs strengthens the reliability of the presented methodology.

However, certain aspects require further clarification or elaboration to enhance the manuscript's impact:

The rationale for selecting specific gene targets and compounds could be expanded to provide deeper biological context. While the results are promising, a more detailed discussion on the limitations of the machine learning models and potential biases in the dataset would strengthen the narrative.

Additional information on the translational implications of the findings, especially for rare diseases, could further highlight the study's significance.

Overall, this manuscript presents a well-constructed study with high potential for impact in the field of computational drug discovery. Minor revisions addressing the points above will enhance its clarity and reach.

**Reviewer #2:**  This paper proposes a pipeline to identify new drug-gene connections by means of Machine Learning techniques applied to the Tox21 in vitro assays. The final objective is to generate a set of repurposing predictions for a list of rare diseases.

Regarding the presentation, the work is clearly organized and easy to follow throughout the document. Authors have made an excellent job in describing the pipeline and synthetizing the analysis without loosing important information.

Technically, some major doubts arose from this paper:

1) The limited descriptive analysis included regarding the used training dataset and test set. It is not clear, for example, how many instances are active or inactive (what I believe to be the positive and negative class respectively in this classification problem).

2) The previous point leads me to this one: the use of AUROC as the main evaluation metric. Have authors considered other metrics as the Area Under the Precision-Recall Curve (AUPRC)? It might be more informative if the problem is not balanced between classes.

3) My main concern is that there is not a code repository linked so I am not able to assess it.

I have also a list of minor concerns that could be addressed:

4) Selection and standardization of rare diseases. Author state that they use OMIM to select the set of rare diseases, but I wonder if standardized sources for rare diseases (such as Orphanet) would have provider a wider list of diseases and their corresponding genes.

5) Figures 4 and 5 lack of the sufficient quality so I am unable to assess them.

6) References should be double checked. In particular, references 49, 53, 54 and 91 seem incomplete.

7) In the abstract, authors claim “with predictions further validated by experimental results” sounds a little bit as they did validate the predictions. Maybe they could reformulate this to they something like they use external experimental dataset to further test their predictions.

8) As for the acronyms format, it would be nice if it was harmonized across the document.

9) In the Methodology section, “3. Validation of novel gene-drug pairs with chemical assay”, it seems like the same content with different words is expressed twice. The first time they are mentioned, Pharos and the Board Drug Repurposing Hub (BDRH) are not referenced but they are the second time.

6. PLOS authors have the option to publish the peer review history of their article (what does this mean? ). If published, this will include your full peer review and any attached files.

**Do you want your identity to be public for this peer review?** For information about this choice, including consent withdrawal, please see our Privacy Policy .

Reviewer #1: No

Reviewer #2: No

---

## [Author Response · Author response to Decision Letter 1]

24 Jan 2025

PONE-D-24-43717

January 22, 2025

Re: Resubmission of manuscript entitled “Novel Target Identification towards Drug Repurposing Based on Biological Activity Profiles”

Dear Editors,

Thank you for the opportunity to revise our manuscript, Novel Target Identification towards Drug Repurposing Based on Biological Activity Profiles. We appreciate the careful review and constructive suggestions from you and the reviewers. Following this letter are point-by-point response to the reviewer comments. Changes made in the manuscript are marked using track changes, we also upload a clean version of revised manuscript. The revision has been developed in consultation with all coauthors, and each author has given approval to the final form of this revision.

Thank you for your consideration!

Kind regards,

Qian Zhu

Reviewer #1:

1) The rationale for selecting specific gene targets and compounds could be expanded to provide deeper biological context. While the results are promising, a more detailed discussion on the limitations of the machine learning models and potential biases in the dataset would strengthen the narrative.

Response: We are grateful for the reviewer’s constructive feedback. Below, we address the comments point by point and describe how we have incorporated them into the revised manuscript.

Comment 1: The rationale for selecting specific gene targets and compounds could be expanded to provide deeper biological context.

Response: We appreciate this insightful suggestion. To address this, we have revised the discussion to provide a detailed rationale for selecting specific gene targets and compounds. Particularly, we elaborated on the biological relevance of these gene targets, focusing on their known roles in disease pathways and potential therapeutic implications. For example, NR3C1 was chosen due to its established association with both common diseases, such as polycystic ovarian syndrome, and rare conditions like Generalized Glucocorticoid Resistance. Similarly, the selection of compounds was guided by their activity profiles derived from the Tox21 dataset and their therapeutic potential, as identified through case studies.

The following section has been added to the Methods:

" The gene targets selected for this study were prioritized based on their significant enrichment in compound activity profiles derived from the Tox21 dataset. This enrichment aligns with their known involvement in key disease pathways, particularly those implicated in rare diseases. For instance, the NR3C1 gene—which codes for the glucocorticoid receptor—has well-documented associations with metabolic and inflammatory pathways, making it a compelling target for drug repurposing. Similarly, the compounds included in this analysis were chosen for their robust activity scores, reflecting their potential to modulate these targets effectively. This strategic selection ensures that our predictive models focus on biologically relevant relationships, maximizing their translational potential. In summary, out of the 737 enriched genes, 143 genes with 6,925 related compounds were chosen to be included in the training set for our model. For each gene target, the number of associated drugs (represented by a value of 1 in the data matrix) ranged from 10 to 223. Conversely, all unassociated drugs with gene targets were marked with a value of 0 in the data matrix. Selected genes are detailed in Table S1. "

Comment 2: While the results are promising, a more detailed discussion on the limitations of the machine learning models and potential biases in the dataset would strengthen the narrative.

Response: We agree that a thorough discussion of model limitations and dataset biases is essential for a balanced narrative. We have expanded the Discussion section to include an in-depth analysis of these aspects. Specifically, we discuss the following:

1. Data Imbalance: The inherent imbalance in the dataset, with a scarcity of positive drug-gene pairs, poses a challenge. While metrics such as AUPRC were employed to address this limitation, it remains a potential source of bias.

2. Predictive Model Limitations: Simpler models like KNN demonstrated lower performance due to their reliance on local data structure, which may not capture complex biological interactions. Conversely, more advanced models, such as SVC and XGB, while providing superior statistical performance, pose challenges in interpreting their results in a biological context due to the complexity of their computational algorithms. This highlights the trade-off between accuracy and interpretability, a critical consideration in translational research.

3. Biological Context: Certain gene targets with limited associated compounds exhibited lower predictability, underscoring the need for more comprehensive datasets to improve model performance.

The revised discussion includes the following:

" Despite the promising results with our models, our study acknowledges several limitations. The imbalance in the dataset, characterized by a disproportionately low number of positive drug-gene pairs, can lead to skewed predictions favoring the majority class. While we employed metrics like AUPRC to mitigate this, future studies would benefit from augmented datasets with more balanced distributions. Additionally, the varying performance of machine learning models highlights the need to balance model complexity and interpretability. Simpler models like KNN underperform due to their inability to capture intricate relationships, while more complex models like XGB, although more accurate, present challenges in biological interpretation due to their computational complexity. Finally, the predictability of certain gene targets is constrained by limited biological context, emphasizing the importance of integrating additional functional data into future analyses."

2) Additional information on the translational implications of the findings, especially for rare diseases, could further highlight the study's significance.

Response: We thank the reviewer for this valuable suggestion, which has helped us better emphasize the significance of our study. In response, we have revised the discussion section and added several paragraphs at the end to highlight the translational implications of our findings, particularly for rare diseases. The revised section is as follows:

In general, the findings of this study hold significant translational potential for addressing the unmet medical needs in rare diseases. By utilizing machine learning models, we identified novel gene-drug associations that provide a foundation for targeted therapeutic development. For instance, the high predictability of gene targets such as NR3C1 and KCNH2, coupled with their relevance to rare diseases like Generalized Glucocorticoid Resistance and Short QT Syndrome, demonstrates the potential to prioritize drug candidates for preclinical and clinical evaluations. Furthermore, the diversity of associated compounds for these targets enables the exploration of multiple therapeutic pathways. This approach is particularly valuable for rare diseases, where limited research funding often constrains the development of targeted treatments. By focusing on well-characterized gene-disease relationships, such as those supported by existing evidence, this study makes it feasible to design clinical studies with higher confidence in the underlying biology. Our methodology also has broader implications for drug repurposing in rare diseases. The ability to predict novel gene-drug relationships and validate them experimentally highlights a systematic pipeline that can be applied to other underexplored diseases. This not only accelerates the identification of promising therapeutic candidates but also provides a scalable framework for addressing the broader challenges of drug discovery in rare disease contexts.

Reviewer #2:

Major concerns:

1) The limited descriptive analysis included regarding the used training dataset and test set. It is not clear, for example, how many instances are active or inactive (what I believe to be the positive and negative class respectively in this classification problem).

Response: We appreciate the reviewer’s feedback regarding the descriptive analysis of the training and test datasets. To address this concern, we have provided the number of associated drugs for all 143 gene targets in Supplementary Table 1 (Top 10 gene targets are shown as examples below). This table explicitly details the number of active instances for each gene target, addressing the question of positive and negative class distribution.

To further enhance clarity, we have revised the methods section of the paper to include additional descriptive details. Specifically, we have added the following sentence. We believe these additions provide a more comprehensive overview of the dataset.

“For each gene target, the number of associated drugs (represented by a value of 1 in the data matrix) ranged from 10 to 223. Conversely, all unassociated drugs with gene targets were marked with a value of 0 in the data matrix. Selected genes are detailed in Table S1.”

Gene Number of associated compounds

AR 223

CYP2D6 158

KCNH2 127

HTR2A 117

SI 113

DRD2 101

HRH1 100

CHRM1 98

SLC6A4 97

CYP1A2 97

2) The previous point leads me to this one: the use of AUROC as the main evaluation metric. Have authors considered other metrics as the Area Under the Precision-Recall Curve (AUPRC)? It might be more informative if the problem is not balanced between classes.

Response: We appreciate the reviewer’s insightful suggestion regarding the use of the Area Under the Precision-Recall Curve (AUPRC) as an evaluation metric. The reviewer raises an excellent point about its relevance for imbalanced classification problems, such as ours.

In our study, we initially focused on the ROC-AUC metric as it was used in our previous study with the Tox21 dataset (reference 30 in the paper). However, we agree that incorporating AUPRC provides an additional perspective and allows us to further evaluate the performance of the models under class imbalance.

To address this, we calculated the AUPRC for the four machine learning algorithms (KNN, SVC, RF, and XGB). The results are now included in Table 3 in the revised manuscript. These results show that SVC, RF, and XGB achieved high AUPRC values, further supporting their robustness and indicating that these models do not bias on the majority class to achieve high performance. However, the KNN model demonstrated relatively lower AUPRC scores, suggesting a tendency to favor true positives in its predictions. This behavior is likely due to the simplicity of the KNN algorithm, which relies on identifying the closest data points in the training set and involves minimal model building and hyperparameter tuning.

We have incorporated the calculation of AUPRC into the methods section and included these results, along with a discussion, in the results section. We are grateful to the reviewer for this suggestion, as it has helped us gain a deeper understanding of our models and has improved the rigor of our analysis.

Table 3. Selected Best Parameters

Models Parameters Mean ROC_AUC * Mean AUPRC *

KNN n_neighbours:33, p:3 0.71268 0.50421

SVC c:20, gamma:0.5, kernel: rbf 0.77428 0.80494

RF bootstrap: false; max_depth: 6, max_features:auto, min_samples_leaf: 2, min_samples_split:2, n_estimators: 30 0.75529 0.73537

XGB Colsample_bytree: 0.6, max_depth: 5, min_child_weight: 3, reg_alpha: 1, subsample: 0.8 0.77504 0.73602

3) My main concern is that there is not a code repository linked so I am not able to assess it.

Response: We apologize for not initially including the code repository. The code is available at the following GitHub link: https://github.com/ncats/drug_rep/tree/main/Tox21_Prediction/TargetPrediction/Code_DrugRepurposing

Minor concerns:

4) Selection and standardization of rare diseases. Author state that they use OMIM to select the set of rare diseases, but I wonder if standardized sources for rare diseases (such as Orphanet) would have provider a wider list of diseases and their corresponding genes.

Response: We thank the reviewer for their thoughtful suggestion regarding the use of standardized sources like Orphanet for rare diseases. While Orphanet is a valuable resource, we deliberately chose OMIM to align better with the specific objectives of our study. OMIM provides several key advantages for linking gene targets to rare diseases in our research:

a) Granularity and Specificity of Gene-Disease Associations:

OMIM provides detailed, gene-specific information and establishes precise associations between genes and rare diseases. In contrast, Orphanet often groups diseases into broader categories, which can lead to less specificity when linking a gene to an individual rare disease. For instance, the gene CALM1 is associated with ORPHA:101016 Familial Long QT Syndrome in Orphanet—a broader category compared to the more specific association of OMIM:616247 Long QT Syndrome, 14 in OMIM. This specificity in OMIM enabled us to establish more targeted and actionable gene-disease connections, which is critical for identifying precise therapeutic targets in drug repurposing efforts.

b) Relevance to Genetic Research:

OMIM’s emphasis on genetic and molecular data aligns closely with our study's objective of linking gene targets to diseases. It offers in-depth insights into the genetic basis of diseases, enabling us to accurately map the relationships between genes and rare diseases derived from our predictive models.

c) Completeness of Rare Disease Gene Coverage:

While Orphanet provides valuable epidemiological and clinical information, OMIM is more comprehensive in its coverage of genetic variants and their phenotypic effects, particularly for rare diseases with well-characterized molecular underpinnings.

While OMIM’s structured genetic data aligns closely with our study's focus on hypothesis-driven drug discovery and provides direct, curated linkages critical for drug repurposing, considering the reviewer’s valuable suggestion, we revisited our findings using the Orphanet database to ensure a thorough evaluation. This review confirmed that no additional gene-rare disease pairs were identified beyond the 34 included in our study. To enhance completeness and transparency, we have incorporated the corresponding OrphaCodes for these pairs, where available, and reflected these updates in the revised manuscript.

5) Figures 4 and 5 lack of the sufficient quality so I am unable to assess them.

Response: We appreciate the reviewer’s feedback regarding the quality of Figures 4 and 5. Unfortunately, the ARAX website (one component of the Translator), which generates these knowledge graphs, does not provide functionality to download or export high-resolution versions. However, the generated graphs can always be accessed at ARAX website (Clicking “Knowledge Graph” under “Output” on the side bar menu. We attached the specific URLs to the caption of Figures 4 and 5.

6) References should be double checked. In particular, references 49, 53, 54 and 91 seem incomplete.

Response: We thank the reviewer for pointing out the incomplete formatting of certain references. We changed the reference style through EndNote to fix those incomplete references. The updated references are included in the revised manuscript. We appreciate the reviewer’s attention to detail in helping us improve the quality of our work.

7) In the abstract, authors claim “with predictions further validated by experimental results” sounds a little bit as they did validate the predictions. Maybe they could reformulate this to they something like they use external experimental dataset to further test their predictions.

Response: We appreciate the reviewer’s observation regarding the phrasing in our abstract, particularly the statement, “with predictions further validated by experimental results,” which may mislead the audience. To address this concern, we have rephrased the statement to clarify that external experimental datasets were used to evaluate our predictions. This updated wording is reflected in the revised manuscript to ensure accurate representation of our methodology. Thank you for

---

## [Decision Letter · Decision Letter 1]

11 Feb 2025

Novel Target Identification towards Drug Repurposing Based on Biological Activity Profiles

PONE-D-24-43717R1

Dear Dr. Zhu,

We’re pleased to inform you that your manuscript has been judged scientifically suitable for publication and will be formally accepted for publication once it meets all outstanding technical requirements.

Kind regards,

Sheikh Arslan Sehgal, PhD

Academic Editor

PLOS ONE

Additional Editor Comments (optional):

Reviewers' comments:

Reviewer's Responses to Questions

**Comments to the Author**

1. If the authors have adequately addressed your comments raised in a previous round of review and you feel that this manuscript is now acceptable for publication, you may indicate that here to bypass the “Comments to the Author” section, enter your conflict of interest statement in the “Confidential to Editor” section, and submit your "Accept" recommendation.

Reviewer #1: All comments have been addressed

Reviewer #2: All comments have been addressed

2. Is the manuscript technically sound, and do the data support the conclusions?

Reviewer #1: Yes

Reviewer #2: Yes

3. Has the statistical analysis been performed appropriately and rigorously? 

Reviewer #1: (No Response)

Reviewer #2: Yes

4. Have the authors made all data underlying the findings in their manuscript fully available?

Reviewer #1: Yes

Reviewer #2: Yes

5. Is the manuscript presented in an intelligible fashion and written in standard English?

Reviewer #1: Yes

Reviewer #2: Yes

6. Review Comments to the Author

Reviewer #1: The concerns have been appropriately addressed. Thus, the manuscript is ready for publication.

The concerns have been appropriately addressed. Thus, the manuscript is ready for publication.

The concerns have been appropriately addressed. Thus, the manuscript is ready for publication.

Reviewer #2: I believe all the necessary revisions have been addressed, and the manuscript is now ready for publication. Congratulations to the authors for their hard work and dedication to this study.

7. PLOS authors have the option to publish the peer review history of their article (what does this mean? ). If published, this will include your full peer review and any attached files.

**Do you want your identity to be public for this peer review?** For information about this choice, including consent withdrawal, please see our Privacy Policy .

Reviewer #1: No

Reviewer #2: No

---

## [Editor Report · Acceptance letter]

PONE-D-24-43717R1

PLOS ONE

Dear Dr. Zhu,

I'm pleased to inform you that your manuscript has been deemed suitable for publication in PLOS ONE. Congratulations! Your manuscript is now being handed over to our production team.

Kind regards,

on behalf of

Dr Sheikh Arslan Sehgal

Academic Editor

PLOS ONE